# Sample-Imagined Generator:
# Efficient Virtual Sample Generation Method for Off-policy Reinforcement Learning with Sparse Rewards

## Abstract

Off-policy reinforcement learning (RL) requires extensive real interaction with environment to gain experience for policy learning, presenting a challenge of low sample efficiency, especially in the condition of sparse rewards. To address this, we propose a Sample-Imagined Generator (SIG), which automatically trains a sample generator during environment interaction and could adaptively generate valuable imagined samples for policy learning. Through SIG, the policy greatly reduced the interaction with the environment during training and achieved comparable or even higher performance with those trained only through real interactions. SIG could be combined with any off-policy RL algorithm. Experiment in 5 continuous control tasks demonstrate that by substituting imagined samples for real ones to supplement the experience pool, SIG accomplishes tasks with significantly less interaction with the environment, notably improving sample efficiency across 10 off-policy reinforcement learning algorithms.

## 1 Introduction

Reinforcement learning (RL) has achieved great success in decision-making tasks (Xu et al., 2018; Loftus et al., 2020; Holcomb et al., 2018). Compared to on-policy RL (Singh et al., 2000; Andrychowicz et al., 2020; Hausknecht et al., 2016), off-policy RL (Fujimoto et al., 2019; Thomas & Brunskill, 2016; Peng et al., 2019) learns the policy through the historical samples, which is more robust when adapting to the environment and becomes a promising way in task learning (Yu et al., 2020). However, extensive interaction with the environment also brings the challenge of low sample efficiency (Buckman et al., 2018). Meanwhile, providing the agent with appropriate dense rewards (Eschmann, 2021; Devidze et al., 2021) is another challenging issue. It is difficult to find an accurate reward function that guides the agent to reach the goal step by step. In the vast majority of current RL research, agents are provided with sparse rewards (Wang et al., 2020; He & Lv, 2023) that are easy to set, namely rewards are given only upon completion of the task. The limited information further leads greater interaction for the agent to successfully complete the task.

This issue can be mitigated by improving the algorithm's target updating or value estimation method to encourage exploration, thereby improving sample efficiency. Prior works have considered several modifications, such as introducing an entropy maximization goal during training through an entropy adjustment mechanism (Haarnoja et al., 2018), and adding random noise to the updated target value (Fujimoto et al., 2018) to incentivize broader exploration by the agent. The value estimation method can be modified in combination with Monte Carlo estimation (Wilcox et al., 2022a) to further improve sample efficiency. In other works, expert data can be provided to the agent, including demonstration data (Nair et al., 2018), and pre-collected large-scale datasets for offline reinforcement learning (Kumar et al., 2020). Although these works exhibit remarkable performance, they necessitate the introduction of demonstrations or meticulously designed hyperparameters, which impedes their application to diverse tasks. Additionally, they cannot actively reduce actual interactions with the environment and still require extensive interaction steps to complete learning.

In this paper, we propose a simple and efficient sample generation method called Sample-Imagined Generator (SIG), which learns from experienced real samples and continuously generates close-

to-real imagined samples for policy learning, greatly improving learning efficiency while actively reducing actual environmental interactions. SIG can be combined with any off-policy RL method as an additional module and easy to implement. The main contributions of this paper are summarized as follows:

• We propose a Self-validating Sample Generator module which introduces an action-validation mechanism to make sample generation in a closed loop. This module could verify the rationality of the imagined states and progressively enhance the state-action mapping relationship, supplying the high-quality imagined samples.

• We propose a Self-adaptive Imagination Inference module which can adaptively adjust the length of the imagined sample trajectory and the quantity incorporated into policy learning, thereby providing the agent with the most appropriate imagined samples at each updating step. This module also actively determines the switch time when the agent can start to reduce interaction, guaranteeing the steady policy convergence during training under reduced environmental interactions.

• We combined SIG with 10 off-policy RL algorithms, chosen from classic methods and recent significant improved methods. To further prove the effectiveness of our method, we selected 5 continuous control tasks with sparse rewards, which makes them more challenging. The empirical results of 50 scenarios show that our method achieved comparable or even better performance compared to the baselines with as little interaction as possible, and greatly improved the sample efficiency of off-policy RL.

## 2 RELATED WORK

The issue of low sample efficiency in off-policy reinforcement learning under sparse rewards can be mitigated by introducing expert data, improving Q-value estimation or policy optimization, which are respectively external and internal improvement methods.

**Introducing Expert Data:** Since the reward signal is sparse, the agent requires more experience gained from exploration to update its policy. One approach to improve sample efficiency is to provide the agent with additional expert experience. A typical type of expert data is demonstration, which led to the development of reinforcement learning from demonstrations (RLfD) (Brys et al., 2015; Gao et al., 2018; Alakuijala et al., 2021). OEFD (Nair et al., 2018) is based on the Deep Deterministic Policy Gradient (DDPG) (Lillicrap et al., 2015) and the Hindsight Experience Replay (HER) (Andrychowicz et al., 2017), demonstrating significant improvements in simulated robotics tasks with limited demonstration data. FLAIR (Chen et al., 2023) combines demonstration data with inverse RL and employs an adaptive mechanism to learn from demonstrations. It can quickly adapt to new demonstrations and make personalized adjustments. RoboCLIP (Sontakke et al., 2024) leverages pre-trained video and language models (Bagad et al., 2023) to generate rewards from human demonstration videos, significantly improving zero-shot performance on robot operation tasks. In other works, large-scale expert datasets can also be used to directly perform offline RL (Levine et al., 2020; Prudencio et al., 2023). CQL (Kumar et al., 2020) addresses the problem of value function overestimation by learning a conservative Q-function and combines expert data for offline learning, which significantly outperforms existing methods.

**Improving Policy Optimization or Q-value Estimation:** Some work focuses on the algorithm's intrinsics, encouraging agents to explore or stabilize the learning process by improving policy optimization (Hu et al., 2023; Zhao et al., 2024), thereby improving sample efficiency. SAC (Haarnoja et al., 2018) maximizes the entropy of the policy, encouraging agents to search for high-reward policies, thereby conducting comprehensive exploration in environments with sparse rewards or high uncertainty. TRPO (Schulman, 2015) defines a trust region to ensure that each policy update does not deviate too far from the original policy, thereby reducing instability in policy learning. POMO (Kwon et al., 2020) exploits symmetry in combinatorial optimization problems and finds all optimal solutions through diversified policy optimization. Other work stabilizes algorithm learning by improving Q-value estimation (Yang et al., 2020; Zhang et al., 2024), better utilizing the current sample smooth policy updating and enhancing sample efficiency. TD3 (Fujimoto et al., 2018) introduces noise disturbance in the Q-value calculation to enhance the robustness of learning. GQE (Schulman et al., 2015) proposes a novel estimation method which significantly reduces the variance of policy gradient estimates. This method exhibits robustness and efficiency when training value functions,

mitigating issues of overfitting. Based on part of demonstrations, MCAC (Wilcox et al., 2022a) proposes a new Q-value estimation method combined with Monte Carlo estimation, enabling sparse rewards to propagate for a long time during the learning process and improving the sample efficiency of off-policy RL.

The above works have performed well for off-policy RL under sparse rewards. However, these methods do not consider interaction issue with the environment, and require numerous interactions to complete related tasks. In this paper, we design an independent enhancement module from a more intuitive perspective of sample provision, greatly reducing actual interaction with the environment and further improving sample efficiency.

## 3 PRELIMINARIES

Our work builds on standard reinforcement learning that can be considered as a Markov Decision Process (MDP)$\{\mathcal{S}, \mathcal{A}, r, p, \gamma\}$. $\mathcal{S}$ and $\mathcal{A}$ represent the state and action spaces, $r$ is the reward from the environment, $p$ is the transition function and $\gamma$ is the discount factor. The agent observes $s_t \in \mathcal{S}$ at a certain time, samples action $a_t \in \mathcal{A}$ from policy $\pi$, receives reward $r$, and obtains the next state observation $s_{t+1} \sim p(\cdot|s_t, a_t)$. The agent continuously optimizes its policy $\pi$ through rewards $r$ obtained through environmental feedback, until the agent can learn the optimal policy $\pi^*$, making optimal actions in any $s_t \in S$ to maximize returns.

$$\pi^* = \arg\max \mathbb{E}_{\tau \sim \pi} \left[ \sum_{t=0}^{T} \gamma^t r\left(s_t, a_t\right) \right] \tag{1}$$

where $\tau = (s_0, a_0, s_1, a_1, ... s_T)$ and $\tau \sim \pi$ represents $\tau$ is the trajectory obtained by iteration through $\pi$ in MDP.

Off-policy RL temporarily saves the transitions $(s_t, a_t, r_t, s_{t+1})$ from the interation and learns the optimal policy $\pi^*$ from them, which is called experience replay (Schaul et al., 2015). Here the most typical off-policy actor-critic (Degris et al., 2012) framework in off-policy RL is introduced. Actor $\pi(a_t|s_t; \theta)$ is responsible for generating policy while critic $Q(s_t, a_t; \omega)$ is for estimating the state-action value function under the current policy. There are many forms of critic update targets and TD target (Sutton, 1988) is the most commonly used. Critic calculates the state-action value $Q(s_{t+1}, a_{t+1}; \omega)$ and the TD target $y_t$:

$$y_t = r_t + \gamma \cdot Q\left(s_{t+1}, a_{t+1}; \omega\right) \tag{2}$$

where $r_t$ is the historical reward, $\gamma$ is the discount factor.

Critic calculates its loss function $\mathcal{L}$ based on TD target, updating the critic's parameters based on the loss function. Actor outputs the action $a_t(s_t; \theta)$ taken in the current state $s_t$, and calculates the objective function $\mathcal{G}$ based on $Q(s_t, a_t; \omega)$ provided by Critic, which are defined as:

$$\mathcal{L} = \frac{1}{2} \left[ y_t - Q\left(s_t, a_t; \omega\right) \right]^2 \qquad \mathcal{G} = \mathbb{E}\left[Q\left(s_t, a_t; \omega\right)\right] \tag{3}$$

Critic and Actor sample historical experience from the Replay Buffer ($\mathcal{RB}$) and updates the parameters respectively based on above functions.

## 4 METHOD

We propose SIG as shown in Figure 1, which consists two parts: Self-validating Sample Generator and Self-adaptive Imagination Inference.

### 4.1 SELF-VALIDATING SAMPLE GENERATOR

The proposed Self-validating Sample Generator (SSG) includes three components: State Imagination Module, Action Validation Module, and Reward Imagination Module as follows.

• **State Imagination Module (SIM)** $\hat{f}(s_{t+1}|s_t, a_t)$ is a neural network which learns from the transitions $(s_t, a_t, r_t, s_{t+1})$ in $\mathcal{RB}$. It inputs $s_t$ and $a_t$ from policy, imagining the next state $\hat{s_{t+1}}$. SIM simulates the environment dynamics and is used to generate the imagined trajectories.

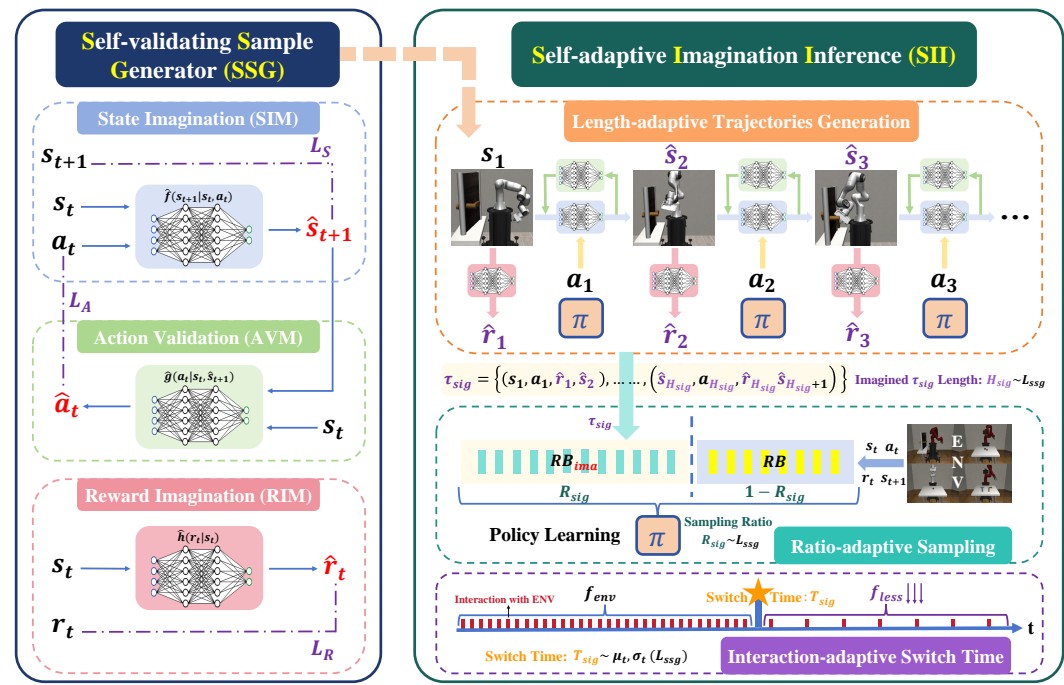

Figure 1: **The structure of SIG**: Left: Self-validating Sample Generator: Virtual samples are generated through State Imagination Module and Reward Imagination Module to accelerate learning, while Action Validation Module forms a closed-loop structure to ensure the high quality imagined samples. Right: Self-adaptive Imagination Inference: $\mathcal{L}_{ssg}$ is used to adaptively adjust the imagined trajectory's length $H_{sig}$ and sampling ratio $R_{sig}$ to provide the agent with the most suitable imagined samples for learning. The appropriate time $T_{sig}$ to start reducing interaction is determined based on the expectation $\mu_t$ and standard deviation $\sigma_t$ of $\mathcal{L}_{ssg}$.

In the actual interaction process, there is noise $\xi$ in the state transfer. And the action $a_t$ follows a certain distribution, which is recorded as $(a_t + \Delta)$, where $\Delta \sim \mathcal{N}(0, \Sigma)$. The actual state transfer can be expressed as:

$$s_{t+1} = f(s_t, a_t) + \xi \tag{4}$$

The loss function $\mathcal{L}_S$ of SIM $\hat{f}(s_{t+1}|s_t, a_t)$ is expressed as:

$$\mathcal{L}_S = ||s_{t+1} - \hat{s_{t+1}}||_2 = ||f(s_t, a_t) + \xi - \hat{f}(s_t, a_t + \Delta)||_2 \tag{5}$$

The deviation caused by $\Delta$ may offset the original $\xi$ in the state transfer process and SIM may learn a wrong state-action relationship that may not meet the actual environment requirements. In order to further ensure the accuracy of SIM and strengthen the correct correlation between the imagined state and action, we introduce an Action Validation Module which forms a self-validation closed loop in SIM.

• **Action Validation Module (AVM)** $\hat{g}(a_t|s_t, \hat{s}_{t+1})$ is another neural network which learns from the transitions in $\mathcal{RB}$ and the outputs of SIM. AVM inputs current $s_t$, next state $\hat{s_{t+1}}$ imagined from SIM, and outputs the predicted actions $\hat{a}_t$. By reducing the discrepancy between the predicted action $\hat{a}_t$ and the actual action $a_t$, the agent's transition prediction becomes more accurate.

The loss function $\mathcal{L}_A$ of AVM $\hat{g}(a_t|s_t, \hat{s}_{t+1})$ is expressed as:

$$\mathcal{L}_A = ||a_t - \hat{a}_t||_2 = ||a_t - \hat{g}(s_t, \hat{f}(s_t, a_t + \Delta))||_2 \tag{6}$$

The loss $\mathcal{L}_{SA}$ of the self-validation closed loop is composed of $\mathcal{L}_S$ and $\mathcal{L}_A$.

$$\mathcal{L}_{SA} = \mathcal{L}_S + \mathcal{L}_A \tag{7}$$

$\mathcal{L}_{SA}$ guides AVM to correct the predicted action $\hat{a}_t$ to be close to the actual action $a_t$, thereby ensuring that $\hat{s}_{t+1}$ is not generated by an unreasonable action deviation $(a_t + \Delta)$. This helps SSG

understand the true state-action mapping relationship of the environment during learning, generating more reasonable and accurate samples, stabilizing and accelerating learning.

• **Reward Imagination Module (RIM)** $\hat{h}(r_t|s_t)$ is also a neural network learning from the transitions in $\mathcal{RB}$. It inputs current state $s_t$ and outputs the imagined reward $\hat{r}_t$. RIM simulates the reward feedback of the environment.

In the sparse reward environment setting, the agent receives a reward of -1 when the task is not completed and a reward of 0 when the task is completed. To robustly simulate this real reward feedback, we perform reward filtering. When $\hat{r}_t$ is near 0 ($[0 - \epsilon_0, 0 + \epsilon_0]$), it indicates that the task is completed at this time, $\hat{r}_t = 0$. When $\hat{r}_t$ is near -1 ($[-1 - \epsilon_0, -1 + \epsilon_0]$), it indicates that the task failed at this time, $\hat{r}_t$ = -1. If $\hat{r}_t$ is in other intervals, it means that the output of RIM is uncertain at this time. We would skip this round of imagination and do not generate any imagined sample, which guarantees the imagined samples are good enough for stable learning.

$$\hat{r}_t = \begin{cases} 0, & \text{if } \hat{r}_t \in [0 - \epsilon_0, 0 + \epsilon_0] \\ -1, & \text{if } \hat{r}_t \in [-1 - \epsilon_0, -1 + \epsilon_0] \\ Null, & \text{otherwise} \end{cases} \tag{8}$$

To ensure more reasonable imagined reward samples, the interval threshold $\epsilon_0$ is set to $10^{-2}$. The loss function $L_R$ of RIM $\hat{h}(r_t|s_t)$ is shown as:

$$\mathcal{L}_R = ||r_t - \hat{r}_t||_2 = ||r_t - \hat{h}(s_t)||_2 \tag{9}$$

The loss $\mathcal{L}_{ssg}$ of SSG is composed of the loss functions of its components, as shown below:

$$\mathcal{L}_{ssg} = \mathcal{L}_{SA} + \mathcal{L}_R \tag{10}$$

## 4.2 Self-adaptive Imagination Inference

SSG is trained with the policy training in parallel. Before it is accurate enough, the imagined samples are not proper to substitute the real samples for policy learning. They would confuse the learning process, even inducing non-convergence. To ensure the valuable imagined samples would be selected for policy training, one way is to self-adaptively adjust the length of the imagined trajectory and the sampling ratio of the imagined samples. So we designed a Self-adaptive Imagination Inference (SII) module to adaptively perform parameter adjustment according to the prediction accuracy of SSG. SII consists of three parts: Length-adaptive Trajectories Generation, Ratio-adaptive Sampling, and Interaction-adaptive Switch Time.

• **Length-adaptive Trajectories Generation** utilizes SSG to provide imagined samples for policy training. It draws on the predictive capabilities of SSG and adaptively adjust the length $H_{sig}$ of imagined trajectory via $L_{ssg}$ as shown below:

$$H_{sig} = h_{env} \cdot (1 - \frac{1}{l_0} \cdot \mathcal{L}_{ssg}) \tag{11}$$

where $h_{env}$ and $l_0$ are the constants decided by the tasks, $h_{env}$ is the maximum step length of the task, $l_0$ is the initial loss of SSG when beginning task policy training.

• **Ratio-adaptive Sampling** decides the number of samples from the imagined trajectories into policy learning. In early stages when SSG is inaccurate, using more real samples for policy training is more conservative. As SSG becomes more accurate, gradually introducing imagined samples would enrich experience and accelerate learning. The sampling ratio $R_{sig}$ is introduced to decide the imagined samples proportion for training at each update iteration:

$$R_{sig} = r_{max} \cdot (1 - \frac{1}{l_0} \cdot \mathcal{L}_{ssg}) \tag{12}$$

where $r_{max}$ is the designated expected maximum imagined sample sampling ratio. The agent samples from $\mathcal{RB}_{ima}$ (the replay buffer saving the imagined samples) and $\mathcal{RB}$ with $R_{sig}$ and $(1 - R_{sig})$ to facilitate policy learning.

• **Interaction-adaptive Switch Time** is the time at which the agent begins to decrease actual interaction with the environment. We expect that when SSG is accurate and stable, actual interactions

will be reduced and more imagined samples would be used for policy training. The switch time prevent premature reduction of interactions leading to insufficient real samples for policy training, causing non-convergence, or delayed reduction of interactions being ineffective as a substitute for the actual environment. The accuracy of SSG is measured by the expectation $\mu_t$ of its loss $\mathcal{L}_{ssg}$, and its stability can be evaluated by the standard deviation $\sigma_t$ calculation in each time as shown below:

$$\sigma_t = \sqrt{\frac{1}{N_t} \sum_{i=1}^{N_t} (\mathcal{L}_{ssg} - \mu_t)^2} \tag{13}$$

where $N_t$ refers to the number of timesteps from the start to the current moment. The switch time $T_{sig}$ is determined adaptively based on $\mu_t$ and $\sigma_t$ at each moment:

$$T_{sig} = \min \{t \mid \forall k \in [t - K + 1, t] \cap \mathbb{Z}, \ \mu(k) < \epsilon \text{ and } \sigma(k) < \tau\} \tag{14}$$

where $\epsilon$ and $\tau$ are the thresholds corresponding to $\mu_t$ and $\sigma_t$ respectively. $K$ is the number of consecutive rounds that meet the threshold condition, which set to a suitable constant of 1000. The moment when the criteria are consecutively fulfilled for $K$ instances is designated as the switch time $T_{sig}$. The agent interacts with the environment at a frequency of $f_{env}$ before $T_{sig}$, and at a reduced frequency of $f_{less}$ after $T_{sig}$. Here $f_{env}$ means normal interaction at every time step ($f_{env} = 1$) and $f_{less}$ means reduced interaction occurring once $\frac{1}{f_{less}}$ steps. This guarantees an adequate sample supply for policy while also proficiently deploying SSG to substitute for environment.

---

**Algorithm 1** The implementation of SIG

---

**Require:** Replay Buffer $\mathcal{RB}$, $\mathcal{RB}_{ima}$, Total episodes $N$
 1: Initialize the replay buffer $\mathcal{RB}$, $\mathcal{RB}_{ima}$
 2: **for** $i \in \{1, \ldots, N\}$ **do**
 3:     **for** $j \in \{1, \ldots, T\}$ **do**
 4:         Sample and execute $a_t^i \sim \pi_\theta(s_j^i)$, observe $s_{j+1}^i, r_j^i$
 5:         $\tau_j^i \leftarrow (s_j^i, a_j^i, r_j^i, s_{j+1}^i)$
 6:         $\mathcal{RB} \leftarrow \mathcal{RB} \cup \{\tau_j^i\}$
 7:         SIG calculates $H_{sig}$ and $R_{sig}$ through $\mathcal{L}_{ssg}$
 8:         SIG generates the imagined trajectories
 9:         $\tau_{sig}^j \leftarrow \{(s_1, a_1, \hat{r}_1, \hat{s}_2), \ldots, (\hat{s}_{H_{sig}}, a_{H_{sig}}, \hat{r}_{H_{sig}}, \hat{s}_{H_{sig}+1})\}$
10:         $\mathcal{RB}_{ima} \leftarrow \mathcal{RB}_{ima} \cup \{\tau_{sig}^j\}$
11:         Sample from $\mathcal{RB}$ and $\mathcal{RB}_{ima}$ with $(1 - R_{sig})$ and $R_{sig}$ for policy updates
12:     **end for**
13:     **if** $\sigma_t < \tau$ and $\mu_t < \epsilon$ **then**
14:         Reduce actual interaction with the environment
15:     **end if**
16: **end for**

---

### 4.3 THE IMPLEMENTATION OF SIG

SIG can be combined with any off-policy RL algorithm to improve learning efficiency while reducing environmental interaction. In the early stages, off-policy RL interacts with the environment to accumulate environmental experience, storing it in $\mathcal{RB}$. SIG continuously trains on these samples to improve SSG's prediction accuracy. SIG utilizes SII to self-adaptively adjust the imagined trajectory length $H_{sig}$, the sampling ratio $R_{sig}$, and the switch time $T_{sig}$.

At each time step SIG randomly selects a sample $s_1$ from $\mathcal{RB}$ and uses this as a starting point for inference, generating imagined trajectory $\tau_{sig} = \{(s_1, a_1, \hat{r}_1, \hat{s}_2), \ldots, (\hat{s}_{H_{sig}}, a_{H_{sig}}, \hat{r}_{H_{sig}}, \hat{s}_{H_{sig}+1})\}$ stored in $\mathcal{RB}_{ima}$ to enhance sample diversity in experience pool. The off-policy RL samples from $\mathcal{RB}$ and $\mathcal{RB}_{ima}$ at ratio $R_{sig}$ for accelerating learning.

When the condition of $T_{sig}$ is satisfied, the frequency $f$ of the agent interacting with the environment reduces from $f_{env}$ to $f_{less}$. At each subsequent time step, more accurate imagined samples progressively replace the real samples, promoting efficient learning and reducing environmental dependence. This results in notable sample efficiency.

## 5 EXPERIMENTS

We combined SIG with 10 off-policy RL algorithms and conducted experiments on 5 continuous control tasks to study and analyze the following issues:

1. The role of SIG in enhancing sample efficiency.

2. The necessity of SIG to replace environmental interactions.

3. The impact of Interaction-adaptive Switch Time on the effectiveness of SIG.

4. The impact of AVM on the effectiveness of SIG.

We summarize the statistics of 10 random seeds for all experiments and report the mean and standard error of their results by exponential smoothing in experimental figures. Our experiments were run on 5 RTX 4090 GPUs with 90GB of memory, 12 vCPU Intel(R) Xeon(R) Platinum 8352V CPU.

**Control Tasks:** 5 continuous control experimental tasks are selected in the study, including Lift (Zhu et al., 2020), Door (Zhu et al., 2020), Extraction (Wilcox et al., 2022b), Push (Thananjeyan et al., 2021), and Navigation (Wilcox et al., 2022a). Each task has a set maximum timestep $T_{task}$ and utilizes **sparse reward** setting. The first four tasks are robot tasks, with the environment featuring a robot equipped with a 7-degree-of-freedom arm and a parallel gripper, as shown in Figure 2.

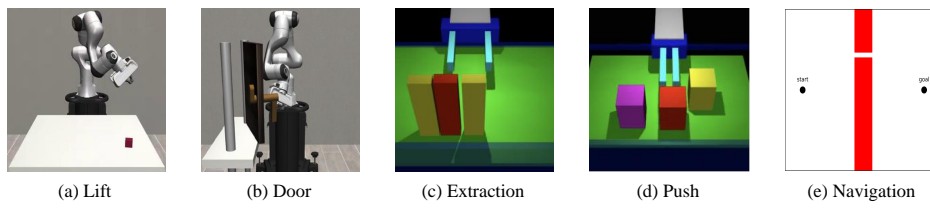

| (a) Lift | (b) Door | (c) Extraction | (d) Push | (e) Navigation |

Figure 2: Five continuous control tasks.

**Off-Policy RL Algorithms:** We choose 5 classic off-policy RL algorithms, These include **SAC** (Haarnoja et al., 2018), **TD3** (Fujimoto et al., 2018), **GQE** (Schulman et al., 2015), **OEFD** (Nair et al., 2018), and **CQL** (Kumar et al., 2020). SAC and TD3 represent classical off-policy RL approaches. GQE enhances training stability based on SAC by incorporating a sophisticated reward estimation. OEFD integrates demonstrations into TD3 to provide behavioral assistance. CQL facilitates the learning of a conservative Q function to prevent value overestimation. To provide a fair comparison with other algorithms, we update CQL online after offline pre-training. In addition, we combine the state-of-the-art MCAC (Wilcox et al., 2022a) improvement module with the above 5 algorithms, and call the combined algorithms **SM** (SAC+MCAC), **TM** (TD3+MCAC), **GM** (GQE+MCAC), **OM** (OEFD+ MCAC) and **CM** (CQL+MCAC), a total of 10 off-policy RL algorithms are selected.

### 5.1 EXPERIMENTAL RESULTS

Experiments were conducted in 5 control tasks to study the role of SIG in improving sample efficiency. For each algorithm, there are two experimental curves: **Algo** and **Algo + SIG**. The solid part of the curve represent the timesteps of the normal interaction with environment and the dotted part represent the reduced interaction. For the curves without dotted part, that means no suitable $T_{sig}$ is identified during training. $f_{less}$ of reduced interaction is set to $4 \cdot 10^{-3}$ in Lift, Door, and Extraction, and $8 \cdot 10^{-3}$ in Push and Navigation, according to the task environment set.

In Figure 3, we study the role of SIG of 10 off-policy RL algorithms and conduct specific analysis based on the number of environmental interaction steps in Appendix C. For classic algorithms such as SAC, TD3, and GQE in Figure 3 (a-e), SAC + SIG in Door and Extraction accelerated learning and reduced the number of interaction steps by **35.05%** and **7.95%**, even completing the challenging Lift task that was difficult for SAC. TD3 + SIG efficiently completed learning in Door by reducing interactions by **61.81%**. GQE + SIG further improved sample efficiency by **18.82%**, **67.18%**, **19.16%**, and **32.28%** in Lift, Door, Extraction, and Navigation, especially having better learning

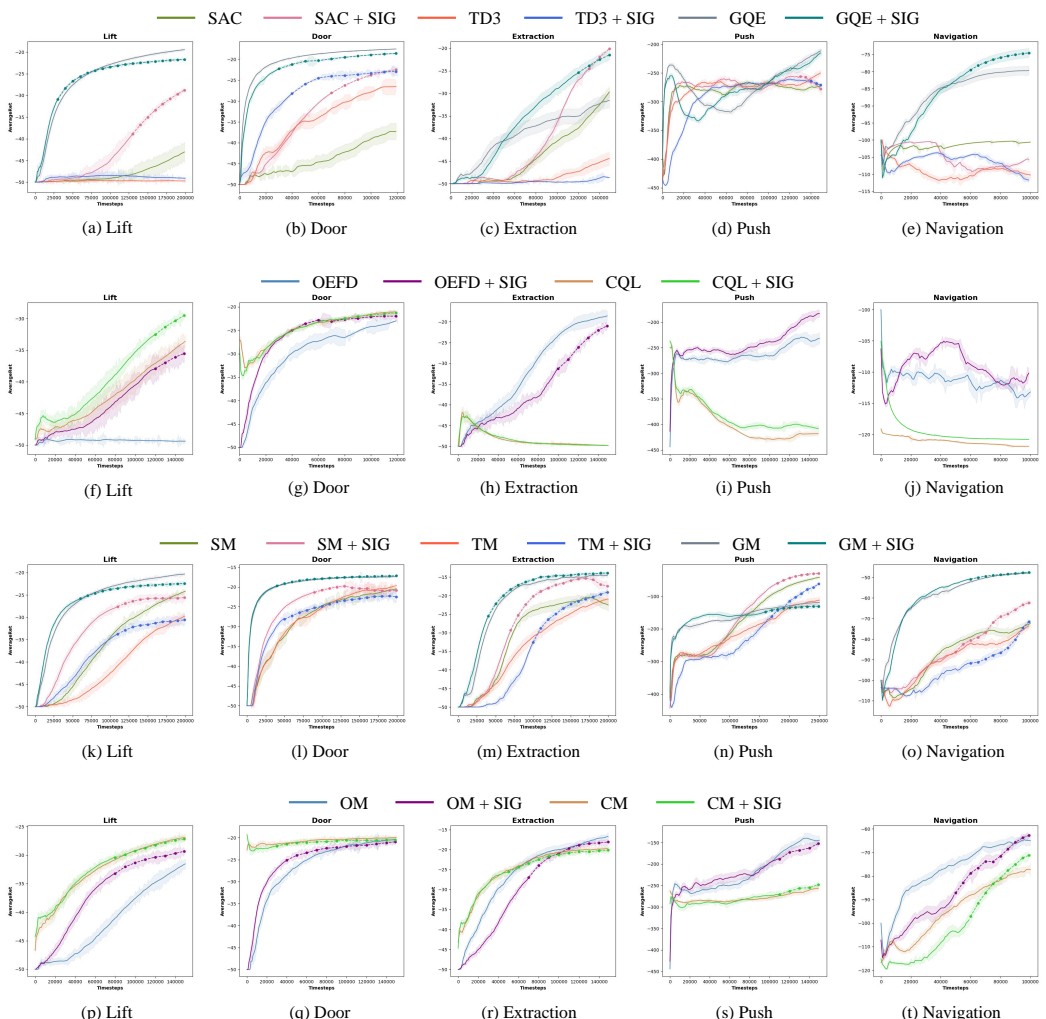

Figure 3: The results of SIG on off-policy RL.

effect on Extraction. Classic methods rely on environmental experience and tend to struggle with task completion or have low efficiency under sparse rewards. SIG addressed this by generating imagined samples, and can even enable task completion in cases where classic methods failed.

OEFD introduces demonstration during experience replay and uses behavioral cloning loss $\mathcal{L}_{BC}$ to further optimize the policy. In Figure 3 (f-j), OEFD + SIG can complete difficult Lift task that the original OEFD failed, and achieved an interaction reduction of **65.56%** and **31.49%** in Door and Extraction. In Push and Navigation tasks, although OEFD + SIG did not find a suitable switch time, it still achieved higher learning efficiency with the assistance of imagined samples. The main reason is that imagined samples generated by SIG are saved in $\mathcal{RB}_{ima}$, and when combined with the demonstration in $\mathcal{RB}$, they can provide more diverse samples for $\mathcal{L}_{BC}$, thereby improving the efficiency of the RLfD methods.

CQL first uses expert data for offline pre-training and then performs online RL interacting with the environment. In Figure 3 (f-j), for the challenging Extraction, Push, and Navigation tasks, the original CQL mainly relied on pre-training rather than environmental samples and had difficulty completing the task. Therefore, the effective imagined samples generated by SIG did not provide significant help. However, CQL + SIG accelerated learning with **15.64%** interaction decrease in Lift and reduced environmental interactions by **25.93%** in Door task, which indicates that the imagined samples from SIG can be combined with expert data in $\mathcal{RB}$ to promote learning for offline pre-trained agents.

MCAC introduces demonstration and Monte Carlo value estimation, which enables the propagation of sparse rewards over long period, thereby improving the learning effect of traditional RL methods. In Figure 3 (k-o), SM + SIG had greater learning speed across 5 tasks, achieving the interaction reduction by **43.09%**, **51.07%**, **61.45%**, **24.86%**, and **47.78%**. TM + SIG gained improved outcomes by decreasing the interactions by **43.38%**, **70.57%**, **44.91%**, **30.01%**, and **34.42%** for each task. Although GM have already possessed excellent performance, GM + SIG achieved comparable results while reducing environmental interactions by **66.83%**, **74.90%**, **76.17%**, **32.68%**, and **39.26%** across all 5 tasks Individually. In Figure 3 (p-t), OM + SIG demonstrated improved sample efficiency across all 5 tasks, reducing interactions by **43.17%**, **77.73%**, **52.82%**, **20.82%**, and **39.88%** respectively. CM + SIG achieved comparable results and reduced interactions by **43.17%**, **83.39%**, **65.42%**, **21.39%**, and **35.76%** individually in 5 tasks. SIG generates imagined samples and incorporates them into policy training, along with MCAC demonstrations. This enriches sample diversity and provides more comprehensive value information for state-action pairs. To further illustrate the necessity of SIG after reducing interaction, we compared our method with **Algo (Less Interact)** (Algo starts to reduce interactions at a fixed frequency ($f_{less} = 4 \cdot 10^{-3}$) when the timestep reaches $\frac{1}{2}T_{task}$). The result of TM + SIG is analysed here (shown in Figure 4) and other groups are in Appendix B.1. We can see that the learning effects of Extraction and Navigation obviously declined after reducing interaction without SIG. The performance of TM + SIG is higher than TM (Less Interact) during learning process in Lift, Door and Push, especially in Lift task. This proves the imagined samples from SIG help to improve the policy learning when reducing interactions.

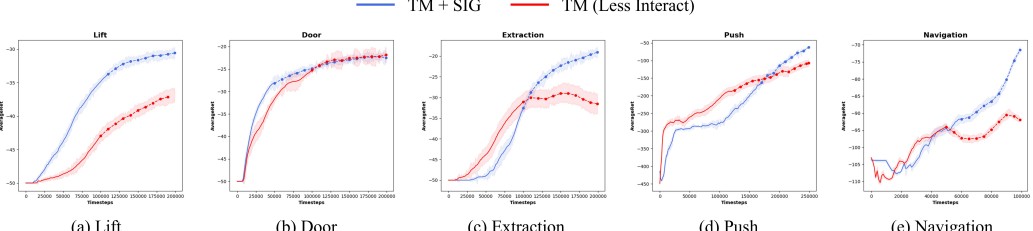

(a) Lift       (b) Door       (c) Extraction       (d) Push       (e) Navigation

Figure 4: The experiment about the necessity of SIG after reducing interaction.

## 5.2 ABLATION EXPERIMENTS

In order to further illustrate the effectiveness and contribution of components in SIG, ablation experiments are performed in 5 continuous control tasks.

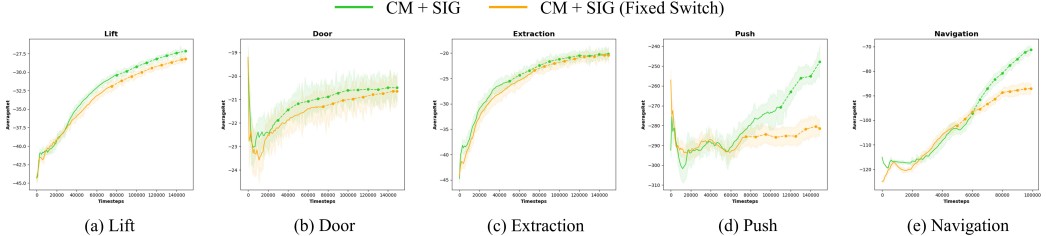

(a) Lift       (b) Door       (c) Extraction       (d) Push       (e) Navigation

Figure 5: The ablation experiment on Interaction-adaptive Switch Time.

To illustrate the effectiveness of the Interaction-adaptive Switch Time, we conducted experiments of **Algo + SIG** and **Algo + SIG (Fixed Switch)** whose switch time is fixed at the timestep of $\frac{1}{2}T_{task}$. To make the comparison reasonable, we choose Algos of SM, TM, GM, OM, and CM shown in Figure 5, because these Algos found $T_{sig}$ during training. The result of CM is analysed here and the other groups are shown in Appendix B.2. We can see that CM + SIG found the optimal switch time to reduce the interactions while sustaining the learning efficiency in all tasks, resulting in comparable or improved learning outcomes than CM + SIG (Fixed Switch). That means SIG can identify the appropriate moment when the imagined samples are good enough to replace the real samples.

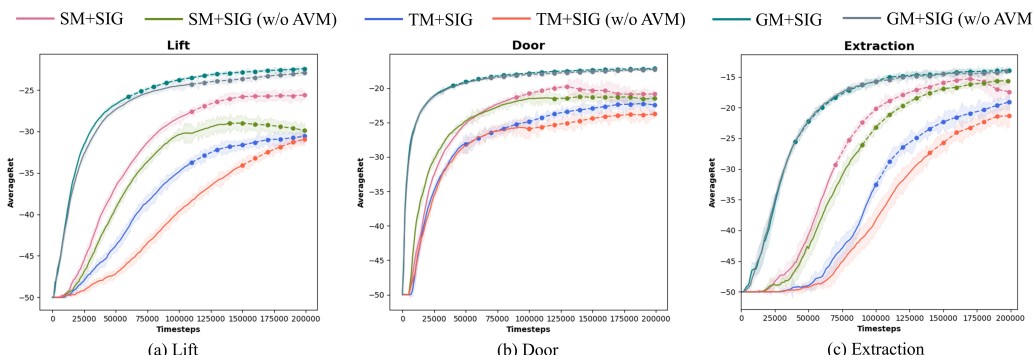

(a) Lift          (b) Door          (c) Extraction

Figure 6: The ablation experiment on AVM.

To demonstrate the effectiveness of AVM, we conducted ablation experiments of SM + SIG, TM + SIG, and GM + SIG in the 3 tasks of Lift, Door, and Extraction, for a total of 9 groups of experiments. Each group was divided into **Algo + SIG** and **Algo + SIG (w/o AVM)** which removes AVM in SIG.

Experimental diagrams are presented in Figure 6. Results indicate that when SIG lacks AVM, due to insufficient state-action understanding, its imagined samples interfere with learning and incur greater loss. SIG had higher learning efficiency and better learning performance. It is concluded that the designed AVM aids in improving the effectiveness of samples imagined by SIG.

## 6 CONCLUSIONS

We propose SIG (Sample-Imagined Generator) which is a sample efficiency enhancer for off-policy RL with sparse rewards. SIG consists two modules including SSG (Self-validating Sample Generator) and SII (Self-adaptive Imagination Inference). SSG generates the high-quality imagined sampled with an action-validation closed loop mechanism. SII adaptively adjusts the length and sample-ratio of imagined trajectories and selects the optimal switch time of reducing interaction according to the accuracy of SSG. The synergy of the two modules could continuously supply the valuable imagined samples for policy training and greatly reduce the real interactions. The empirical results verified the satisfied performance of SIG with 10 off-policy RL algorithms across 50 scenarios and showed the promise of applying RL in real applications. In the future, we would continue to focus on the research of high sample efficiency algorithms for RL and further develop SIG to be more stable and applicable in real-world.

## REPRODUCIBILITY STATEMENT

The code of our method is provided in a zipped file to the supplementary material. We used publicly available RL environments for our 5 simulated tasks as mentioned in Section 5, see Appendix A for more environmental details. The code includes the implementation of task environments and our method, as well as scripts to reproduce the experimental results reported in this paper. All results can be reproduced easily according to the instruction provided in our file.

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

## A    Detailed Experiment Instructions

In this section, we introduce more details of our experiments. The 5 sparse reward task environments used are specifically described below:

**(a) Lift:** In the Lift task, with a horizon of 50, the robot is positioned in front of a table with a single block randomly placed on it. The task requires the robot to lift the block to a specified height. Failure to lift the block incurs a reward of -1, while successfully lifting it yields a reward of 0.

**(b) Door:** In the Door task, with a horizon of 50, the robot is positioned on the floor, and doors at various positions on the table are randomly generated. The objective is for the robot to turn the door handle to open the door. Failure to do so results in a reward of -1, whereas successfully opening the door yields a reward of 0.

**(c) Extraction:** In the Extraction task, with a horizon of 50, the robot is positioned in front of a table with three squares randomly placed on it. The task involves extracting a specific red square from the three squares. Failure to extract the red square incurs a reward of -1 at each timestep, while successfully doing so results in a reward of 0.

**(d) Push:** In the Push task, with a horizon of 150, the robot is positioned in front of a table with three blocks randomly positioned on it. The objective is to sequentially push the blocks forward. Each successful block push increments the reward by 1. Failure to make progress incurs a reward of -3, while completing the task yields a reward of 0.

**(e) Navigation:** In the Navigation task, with a horizon of 100, actions in the environment involve incremental position changes in the x and y directions. The goal is for a point mass to navigate from the starting point to the end point while avoiding obstacles. Collision with an obstacle results in a reward of -100, terminating the episode. Failure to reach the target yields a reward of -1, while successful navigation yields a reward of 0.

In addition, for SIM, AVM, and RIM in SSG, the networks are all designed as that consisting of 3 fully-connected layers with 256 hidden units. They are updated using Adam Optimizer (Da, 2014). And the 3 hyperparameters $r_{max}$, $\epsilon$, and $\tau$ set in the experiment are listed in Table 1.

Table 1: Hyperparameters set in the experiment

| Parameters | Lift | Door | Extraction | Push | Navigation |
|:---:|:---:|:---:|:---:|:---:|:---:|
| $r_{max}$ | 0.25 | 0.25 | 0.125 | 0.1 | 0.125 |
| $\epsilon$ | 0.7 | 0.8 | 0.25 | 0.25 | 0.5 |
| $\tau$ | 0.1 | 0.1 | 0.1 | 0.2 | 0.2 |

## B    Additional Experiments

### B.1    Additional Experiments about the Necessity of SIG

In this section, other experiments about the necessity of SIG performed in SM, GM, OM and CM are analyzed corresponding to Section 5.1.

In Figure 7, contrasting **Algo + SIG** with **Algo (Less Interact)**, under conditions of reduced environmental interactions, for SM as shown in Figure 7 (a-e), the absence of SIG results in poorer performance across all 5 task environments. Without SIG, the Extraction task even declined the learning effect rapidly after reducing interactions. In Figure 7 (f-j), due to GM's inherently strong performance and low reliance on samples, the impact of lacking SIG was minimal. However, a downward trend in learning effect was observed for Extraction. The above results show that typical off-policy RL methods rely on environmental samples to conduct policy learning. After reducing environmental interaction, it is necessary to use SIG to provide close-to-real samples, in order to stabilize policy convergence.

In Figure 7 (k-o), for OM using demonstration data for experience replay, after starting to reduce interactions, although the policy can still converge stably, OM + SIG can perform better than OM

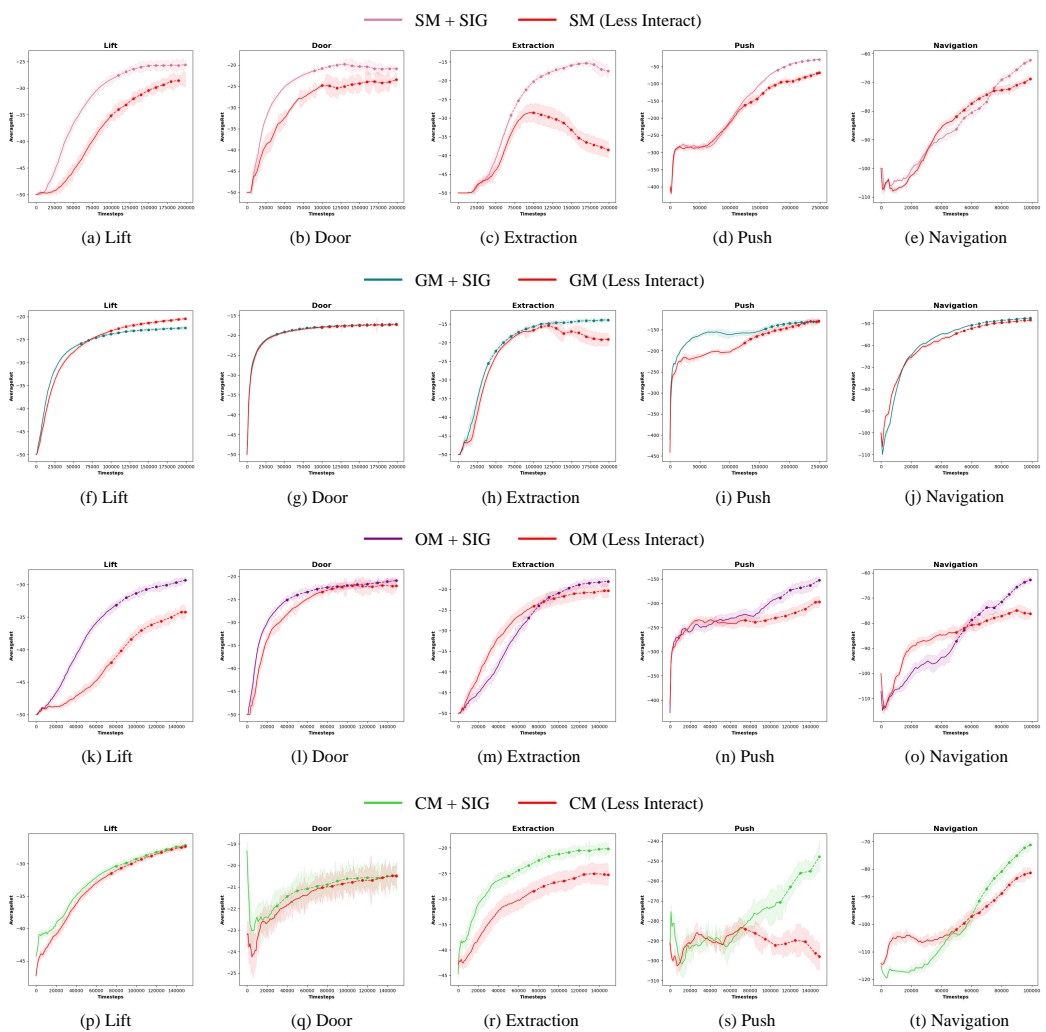

Figure 7: Other experiments about the necessity of SIG after reducing interaction.

in 5 tasks. For RLfD like OM, although it has some ability to resist the reduction of interactions, combining demonstration data with high-quality samples from SIG allows it to better cope with tasks lacking in environmental samples and to perform better.

In Figure 7 (p-t), for CM that uses expert data for offline pre-training, since it mainly relies on expert data rather than environmental samples, it can still maintain a stable convergence policy for Lift, Door, Extraction, and Navigation after reducing interactions. However, for the challenging Push task, due to the lack of valid samples provided by SIG, its learning effect continued to decline until task failure. For offline RL like CM, although it can complete tasks with reduced interactions by leveraging expert data, for some challenging tasks, it is necessary to rely on effective samples provided by SIG for stable learning.

## B.2 ADDITIONAL ABLATION EXPERIMENTS ON INTERACTION-ADAPTIVE SWITCH TIME

In this section, other ablation experiments on Interaction-adaptive Switch Time performed in SM, TM, GM and OM are analyzed corresponding to Section 5.2.

In Figure 8, comparing the performance of **Algo + SIG** with **Algo + SIG (Fixed Switch)**, in the case of SM in Figure 8 (a-e), SM + SIG achieved slightly better learning results in Lift, Door and Navigation tasks. In Figure 8 (f-j), TM + SIG found the suitable time to replace the environment

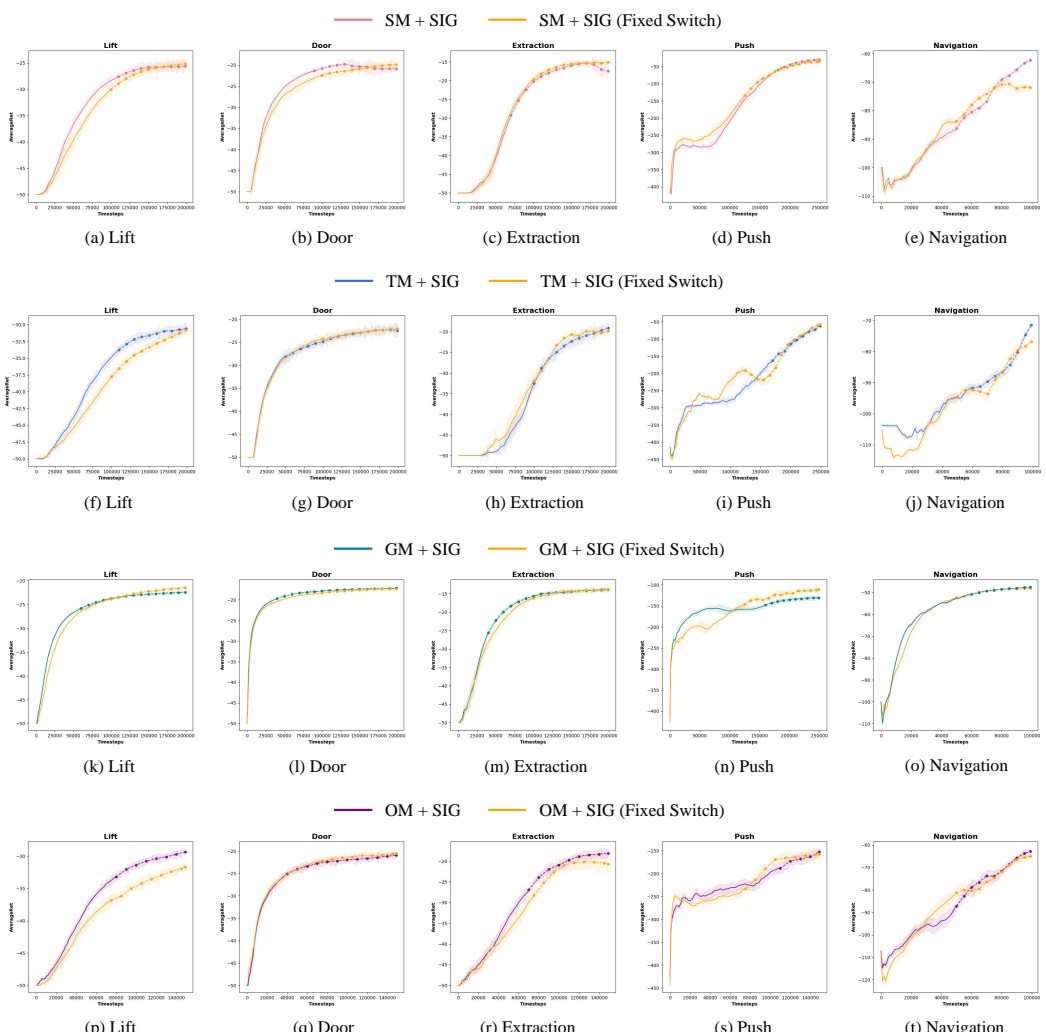

Figure 8: Other ablation experiments on Interaction-adaptive Switch Time.

with SIG, achieving greater performance. For GM in Figure 8 (k-o), GM + SIG yielded results that are comparable to those of the fixed switch in 5 tasks. For OM in Figure 8 (p-t), OM + SIG found an earlier switch time in Extraction, and a later switch time in Lift, both possessing higher learning effects. Experimental results demonstrate that compared with the inflexible fixed switch, the Interaction-adaptive Switch Time adaptively selects the most appropriate switch time according to different tasks, either early or late. The performance of Interaction-adaptive Switch Time is comparable to or better than that of fixed switch, indicating that this adaptive mechanism is effective for applying SIG to different tasks.

## C    THE PERFORMANCE IN REDUCING INTERACTIONS OF SIG

One of SIG's main contributions is determining the appropriate time point to start reducing environmental interactions based on SSG's accuracy, thereby significantly reducing the number of interactions. In this section, we record the specific number of steps of environmental interactions and calculate the reduction ratio, as shown in Table 2 and Figure 9. They also serve as supplementary materials for the experimental analysis in Section 5.1.

Table 2 records the number of steps of 10 off-policy RL algorithms (with or without SIG) interacting with the environment under 5 tasks in the form of data, and calculates the reduction ratio of envi-

Table 2: Number of steps to interact with the environment

| Algorithms | Lift | Door | Extraction | Push | Navigation |
|---|---|---|---|---|---|
| SAC | 200000 | 120000 | 150000 | 150000 | 100000 |
| SAC + SIG | 138271 | 77937 | 138075 | 139548 | 100000 |
| $\delta$ | N/A | **35.05%** | **7.95%** | **6.97%** | N/A |
| TD3 | 200000 | 120000 | 150000 | 150000 | 100000 |
| TD3 + SIG | 200000 | 45832 | 150000 | 145559 | 100000 |
| $\delta$ | N/A | **61.81%** | N/A | 2.96% | N/A |
| GQE | 200000 | 120000 | 150000 | 150000 | 100000 |
| GQE + SIG | 65845 | 39379 | 121258 | 150000 | 67220 |
| $\delta$ | **18.82%** | **67.18%** | **19.16%** | 0.00% | **32.28%** |
| OEFD | 150000 | 120000 | 150000 | 150000 | 100000 |
| OEFD + SIG | 121766 | 41324 | 102768 | 150000 | 100000 |
| $\delta$ | N/A | **65.56%** | **31.49%** | 0.00% | 0.00% |
| CQL | 150000 | 120000 | 150000 | 150000 | 100000 |
| CQL + SIG | 126533 | 88889 | 150000 | 150000 | 100000 |
| $\delta$ | **15.64%** | **25.93%** | N/A | N/A | N/A |
| SM | 200000 | 200000 | 200000 | 250000 | 100000 |
| SM + SIG | 113815 | 97855 | 77096 | 187853 | 52220 |
| $\delta$ | **43.09%** | **51.07%** | **61.45%** | **24.86%** | **47.78%** |
| TM | 200000 | 200000 | 200000 | 250000 | 100000 |
| TM + SIG | 113242 | 58853 | 110177 | 174968 | 65584 |
| $\delta$ | **43.38%** | **70.57%** | **44.91%** | **30.01%** | **34.42%** |
| GM | 200000 | 200000 | 200000 | 250000 | 100000 |
| GM + SIG | 66344 | 50197 | 47663 | 168305 | 60741 |
| $\delta$ | **66.83%** | **74.90%** | **76.17%** | **32.68%** | **39.26%** |
| OM | 150000 | 150000 | 150000 | 150000 | 100000 |
| OM + SIG | 85243 | 44545 | 70766 | 118769 | 60125 |
| $\delta$ | **43.17%** | **77.73%** | **52.82%** | **20.82%** | **39.88%** |
| CM | 150000 | 150000 | 150000 | 150000 | 100000 |
| CM + SIG | 85242 | 33225 | 51872 | 117922 | 64225 |
| $\delta$ | **43.17%** | **83.39%** | **65.42%** | **21.39%** | **35.76%** |

ronmental interaction steps, clearly explaining the sample efficiency improvement of SIG. The main text mentioned that the frequency of normal interaction is $f_{env}$, the frequency of reduced interaction is $f_{less}$, the set maximum timestep of each task is $T_{task}$, and the interaction-adaptive switch time is $T_{sig}$. The original number of environmental interaction steps $\kappa_0$ is:

$$\kappa_0 = T_{task} \cdot f_{env} \tag{15}$$

With SIG, the number of reduced interaction steps $\kappa_1$ is:

$$\kappa_1 = T_{sig} \cdot f_{env} + (T_{task} - T_{sig}) \cdot f_{less} \tag{16}$$

Therefore, the reduction ratio $\delta$ of environmental interaction can be calculated as:

$$\delta = \frac{\kappa_0 - \kappa_1}{\kappa_0} = \frac{(T_{task} - T_{sig}) \cdot (f_{env} - f_{less})}{T_{task} \cdot f_{env}} \tag{17}$$

The calculation of the reduction ratio $\delta$ is valid only when both Algo and Algo + SIG are successful. If the original method Algo in Table 2 fails within $T_{task}$, the calculation of the corresponding $\delta$ is invalid and recorded as N/A. Some Algo + SIG did not find a suitable switch time $T_{sig}$ during the whole process, so environmental interaction was not reduced, and $\delta$ is 0.00%.

In Figure 9, the histogram vividly shows that SIG greatly reduces the actual interaction with the environment. The sub-figures in the left column correspond to the five algorithms SAC, TD3, GQE,

OEFD and CQL, and sub-figures in the right column correspond to the five algorithms SM, TM, GM, OM and CM. Combined with the experimental results in Section 5.1, it illustrates that SIG can help off-policy RL efficiently complete policy learning while greatly reducing interaction, significantly improving sample efficiency.

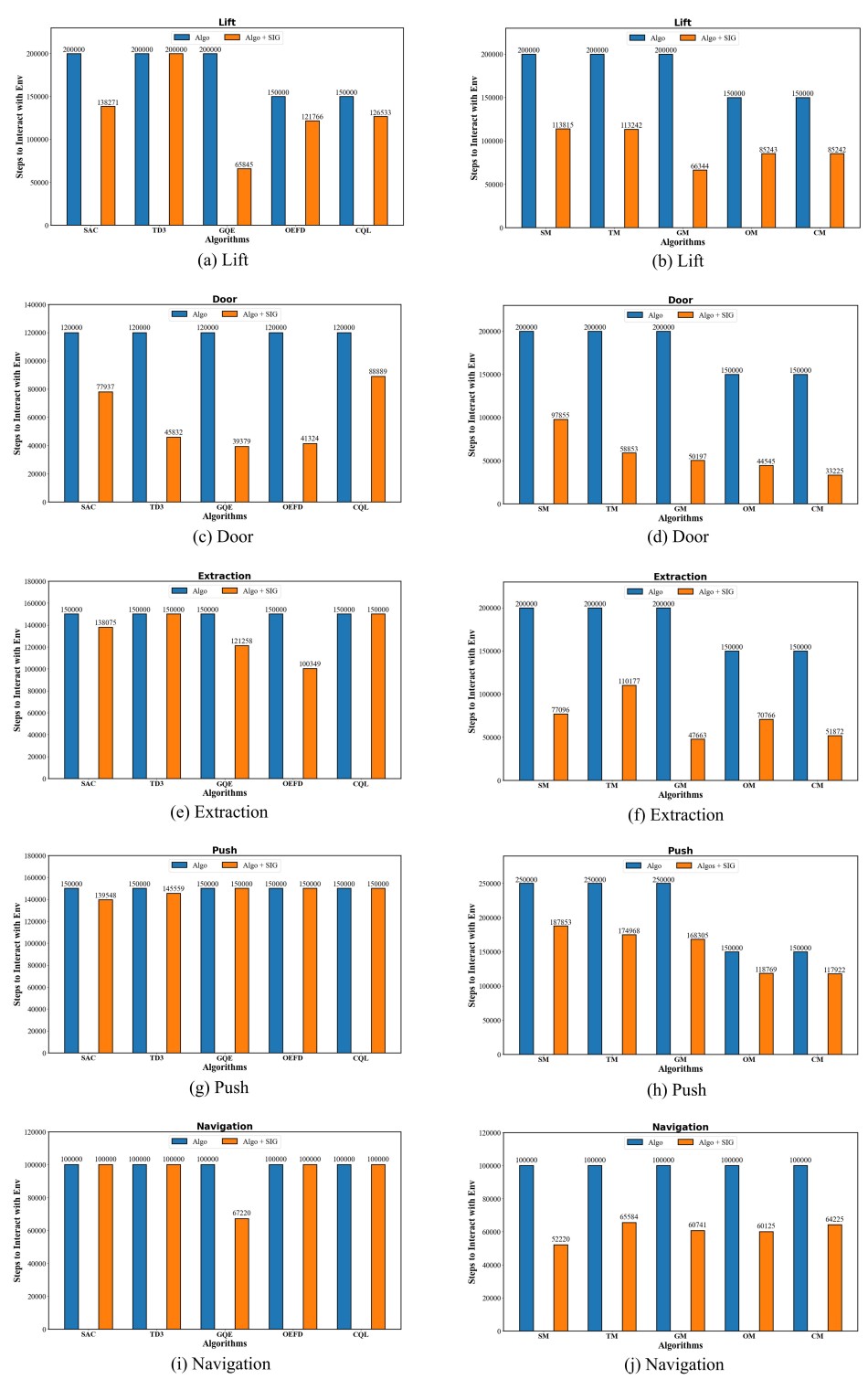

Figure 9: Histogram of environmental interaction steps.

## D  THE OVERALL CHANGES IN SSG'S LOSS $\mathcal{L}_{ssg}$

SII ensures that SIG could provide high-quality samples for the agent through the adaptive parameters adjustment based on $\mathcal{L}_{ssg}$, which makes $\mathcal{L}_{ssg}$ become a key variable in the imagined sample generation process. In this section, we record the overall changes of $\mathcal{L}_{ssg}$ under different tasks as supplementary material for additional reference.

We record the changes in $\mathcal{L}_{ssg}$ and its standard deviation $\sigma_t$ when running the SAC + SIG algorithm in 5 tasks: Lift, Door, Extraction, Push, and Navigation. When the agent explores a new area and the data distribution changes, $\mathcal{L}_{ssg}$ will suddenly increase. We selected the moment of sudden loss increase and 20 points around it for each task and recorded their loss values in Table 3 below.

Table 3: The Changes of SSG's loss value $\mathcal{L}_{ssg}$ in 5 Tasks

| Epoch | 1 | ... | 50 | 51 | 52 | 53 | 54 | 55 | 56 | 57 | 58 | 59 | 60 |
|---|---|---|---|---|---|---|---|---|---|---|---|---|---|
| Lift | 0.85 | ... | 0.64 | 0.64 | 0.66 | 0.67 | 0.65 | 0.64 | 0.64 | 0.67 | 0.67 | 0.67 | 0.73 |
| Epoch | 61 | 62 | 63 | 64 | 65 | 66 | 67 | 68 | 69 | 70 | ... | 200 | |
| Lift | 0.67 | 0.63 | 0.67 | 0.65 | 0.65 | 0.62 | 0.63 | 0.67 | 0.66 | 0.63 | ... | 0.53 | |
| Epoch | 1 | ... | 99 | 100 | 101 | 102 | 103 | 104 | 105 | 106 | 107 | 108 | 109 |
| Door | 0.93 | ... | 0.62 | 0.64 | 0.65 | 0.58 | 0.61 | 0.60 | 0.62 | 0.63 | 0.61 | 0.66 | 0.68 |
| Epoch | 110 | 111 | 112 | 113 | 114 | 115 | 116 | 117 | 118 | 119 | ... | 200 | |
| Door | 0.60 | 0.60 | 0.62 | 0.60 | 0.60 | 0.59 | 0.57 | 0.60 | 0.60 | 0.61 | ... | 0.56 | |
| Epoch | 1 | ... | 66 | 67 | 68 | 69 | 70 | 71 | 72 | 73 | 74 | 75 | 76 |
| Extract | 0.45 | ... | 0.27 | 0.26 | 0.27 | 0.29 | 0.29 | 0.24 | 0.29 | 0.26 | 0.27 | 0.24 | 0.34 |
| Epoch | 77 | 78 | 79 | 80 | 81 | 82 | 83 | 84 | 85 | 86 | ... | 200 | |
| Extract | 0.25 | 0.23 | 0.25 | 0.24 | 0.23 | 0.23 | 0.27 | 0.24 | 0.23 | 0.21 | ... | 0.17 | |
| Epoch | 1 | ... | 37 | 38 | 39 | 40 | 41 | 42 | 43 | 44 | 45 | 46 | 47 |
| Push | 0.76 | ... | 0.51 | 0.43 | 0.53 | 0.47 | 0.48 | 0.52 | 0.44 | 0.47 | 0.47 | 0.49 | 0.82 |
| Epoch | 48 | 49 | 50 | 51 | 52 | 53 | 54 | 55 | 56 | 57 | ... | 250 | |
| Push | 0.26 | 0.41 | 0.45 | 0.40 | 0.34 | 0.59 | 0.60 | 0.34 | 0.59 | 0.44 | ... | 0.22 | |
| Epoch | 1 | ... | 22 | 23 | 24 | 25 | 26 | 27 | 28 | 29 | 30 | 31 | 32 |
| Nav | 0.58 | ... | 0.35 | 0.36 | 0.36 | 0.37 | 0.34 | 0.33 | 0.35 | 0.37 | 0.39 | 0.40 | 0.46 |
| Epoch | 33 | 34 | 35 | 36 | 37 | 38 | 39 | 40 | 41 | 42 | ... | 100 | |
| Nav | 0.44 | 0.41 | 0.40 | 0.38 | 0.37 | 0.36 | 0.35 | 0.34 | 0.34 | 0.32 | ... | 0.12 | |

In order to present the change process of $\mathcal{L}_{ssg}$ in the whole process, we plot the change curve of $\mathcal{L}_{ssg}$ and its standard deviation $\sigma_t$ in the Extraction task, see Figure 10. The red circle in Figure 10 (a) indicates when data distribution is altered, causing $\mathcal{L}_{ssg}$ to rise within a small ascending range initially but drop quickly afterwards.

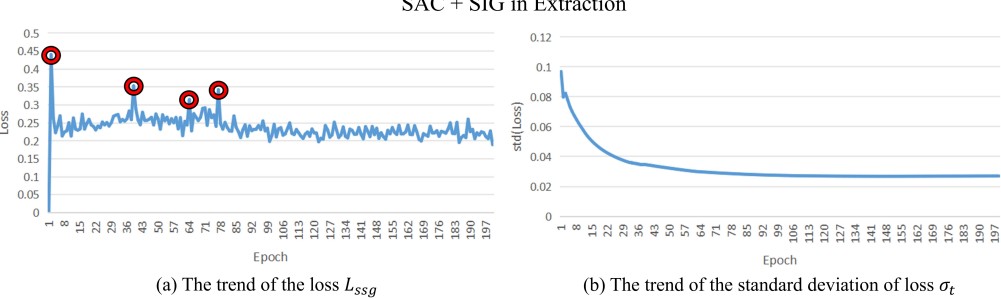

(a) The trend of the loss $L_{ssg}$  (b) The trend of the standard deviation of loss $\sigma_t$

Figure 10: Changes in loss and its standard deviation when running SAC+SIG in the Extraction environment.

SSG's loss $\mathcal{L}_{ssg}$ and its standard deviation $\sigma_t$ exhibit a downward trend throughout the process. Even when the data distribution changes, there is a sudden increase in $\mathcal{L}_{ssg}$, but this surge is within a small range and quickly decreases, converging to a reasonable range. With SSG training, $\mathcal{L}_{ssg}$ gradually converges in a downward trend, providing an effective reference for adaptive parameter adjustment of SII.

