# OpenReview forum: "Sample-Imagined Generator: Efficient Virtual Sample Generation Method for Off-policy Reinforcement Learning with Sparse Rewards"
_ICLR.cc/2025/Conference — Submitted to ICLR 2025_

### Official Review · Reviewer_jEvo · 2024-11-03

**Soundness:** 1
**Presentation:** 2
**Contribution:** 1
**Rating:** 3
**Confidence:** 4

**Summary:**

This paper aims to improve the sample efficiency of off-policy RL algorithms by developing a method called Sample-Imagined Generator (SIG), which generates synthetic rollouts with adaptively varying lengths to replace interactions in the environment with imaginary ones. SIG has two modules: 1) Self-validating Sample Generator module generates imaginary transitions, i.e., states and rewards, and stabilizes imagination through an action validation component. 2) The Self-adaptive Imagination Inference module adaptively adjusts the length of the imaginary rollouts, the ratio of imaginary-to-real samples, and the time to switch from real to imaginary samples. The authors evaluate SIG in 5 continuous control domains with sparse rewards by combining it with ten different off-policy RL algorithms to analyze the benefits of SIG.

**Strengths:**

- The experimental set-up covers many complex continuous control tasks with sparse reward.

- The results show that plugging SIG into specific off-policy RL algorithms can improve sample efficiency.

**Weaknesses:**

- The paper is not well-written. Section 4 is very hard to follow. The authors introduce a lot of modules and components with similar, sometimes confusing names, such as 'Length-adaptive Trajectories Generation.'

- The introduction talks about some existing approaches requiring 'meticulously designed hyperparameters,' yet SIG trains three modules with many hyperparameters: learning rate, network size/depth, covariance on noise, the intervals defined for the reward imagination module, etc. I believe this argumentation is neither fair nor provides a clear motivation for the proposed approach.

- The circular definition in Equation (8) is confusing.

- There are typos: In a lot of places, the hat on a s or r is not placed the letter but the whole character, such as \hat{s_{t+1}} instead of \hat{s}_{t+1}.

- Font sizes of labels/titles on figures are tiny, hence hard to read.

- Quantitative results do not indicate a clear sample-efficiency benefit of using SIG with an off-policy RL algorithm. SIG usually achieves similar performance, but sometimes it seems to be slower.

- The paper does not include qualitative results relating to how imagination works or improves during training.

**Questions:**

- Do the authors consider SIG to be a model-based RL approach?

- Although length-adaptive and ratio-adaptive sampling is clear, the interaction-adaptive switch time is not so much. Is this the time when buffer augmentation through imagination starts, in the sense that until then, no imagination happens during training?

---

### Official Review · Reviewer_87ka · 2024-11-04

**Soundness:** 2
**Presentation:** 1
**Contribution:** 2
**Rating:** 3
**Confidence:** 4

**Summary:**

The paper introduces the Sample-Imagined Generator (SIG), a method designed to improve sample efficiency in off-policy reinforcement learning (RL) with sparse rewards. SIG consists of two main components:
- Self-validating Sample Generator (SSG): Generates high-quality imagined samples using three modules:
State Imagination Module (SIM)
Action Validation Module (AVM)
Reward Imagination Module (RIM)
- Self-adaptive Imagination Inference (SII): Adaptively adjusts the length of imagined sample trajectories and the quantity used in policy learning.

The authors claim SIG can be combined with various off-policy RL algorithms and demonstrate improved sample efficiency across 10 different methods in 5 continuous control tasks with sparse rewards.

**Strengths:**

- Compatibility: The method is designed to work with various off-policy RL algorithms, increasing its potential impact and applicability.
- Self-validating mechanism: The closed-loop structure of the SSG, including the Action Validation Module, aims to ensure high-quality imagined samples.
- Adaptive sampling: The SII module's ability to adjust imagined trajectory length and sampling ratio could potentially optimize the use of imagined samples during training.
- Reduced environmental interactions: SIG aims to achieve comparable or better performance with fewer real environment interactions, which could be valuable in scenarios where interactions are costly or limited.

**Weaknesses:**

- Lack of comparison with model-based RL: SIG bears similarities to model-based RL approaches, but the paper fails to compare it with existing model-based methods, leaving its novelty and effectiveness in context unclear.
- Limited exploration of high-dimensional state spaces: The paper does not address how SIG performs with high-dimensional state spaces, which could be a significant limitation for real-world applications. Generating new states with high-dimensional state spaces (like images) is much more difficult.
- Insufficient experimental results: While the authors claim to have tested SIG with 10 off-policy RL algorithms across 5 continuous control tasks, this range of experiments is still relatively narrow and may not fully demonstrate the method's robustness and generalizability.
- Poor presentation: The paper suffers from unclear writing and organization, making it difficult for readers to follow the proposed method and understand its contributions.
- Lack of computational analysis: The paper does not discuss the computational overhead of implementing SIG, which could be significant due to the additional neural networks and sample generation processes.

**Questions:**

- How does SIG compare to state-of-the-art model-based RL methods in terms of performance and sample efficiency?
- Can you provide results demonstrating SIG's effectiveness in environments with high-dimensional state spaces?
- What is the computational overhead of implementing SIG compared to standard off-policy RL methods?
- How does the performance of SIG degrade as the complexity of the environment increases?
- What are the limitations of SIG, and in what types of environments or tasks might it not be suitable?

---

### Official Review · Reviewer_BYb2 · 2024-11-04

**Soundness:** 2
**Presentation:** 2
**Contribution:** 2
**Rating:** 3
**Confidence:** 3

**Summary:**

The paper introduces the Sample-Imagined Generator (SIG), a framework for generating new experience form a learned model to improve the sample efficiency of RL. Empirically, SIG improves the sample efficiency of a wide range of RL algorithms in 5 complex tasks.

**Strengths:**

1. Model-based data generation techniques are of great interest to the RL community, making this paper quite relevant.
1. SIG is paired with a wide range of RL algorithms in the empirical evaluation.

**Weaknesses:**

I lean to reject primarily because (1) SIG is evaluated across a wide range of RL algorithms but is not compared to any model-based RL baselines such as MBPO [1]
 and PETS [2], and (2) Length-adaptive Trajectories Generation, Ratio-adaptive
Sampling, and Interaction-adaptive Switch Time seem unmotivated and their effect on performance is not ablated. I would like to see ablations on each of these components to understand how important they are for improving sample efficiency.

* Since SIG is not compared to other model-based imagination algorithm, it's difficult to assess whether SIG is improving over existing methods in at least one aspect;

* Eq. 11 essentially says that model rollouts are very short when the SSG loss is large ($L_{ssg}/l_0 \approx 1$ implies that $H_{sig} \approx 0$) and that model rollouts correspond to full trajectories when the SSG loss is 0. However, this does not seem well-motivated; the MBPO paper [1] provides theoretical justification for using shorter model rollouts.  Is there empirical evidence motivating the use of longer model rollouts? I'm wondering if performance would improve if exclusively shorter rollouts were used. In other words, does length-adaptation actually improve performance? MBPO also implements a similar linear increase in the trajectory length over the course of training (e.g see appendix C in [1]). How does SIG's length adaptation relate to MBPO's?

* Eq. 12 SIG integrates more model data into learning as the SSG loss decreases. MBPO keeps this quantity fixed throughout training (e.g. 400 model samples generated per environment step). Is there a benefit to linearly increasing the number of model samples generated vs. keeping it fixed?

* SIG improves sample efficiency for the following setups:
  * SAC: Lift, Door, Extraction (3/5 tasks)
  * TD3: Lift (though TD3 + SIG doesn't seem to solve the task), Door, Extraction  (3/5 tasks)
  * OEFD: Lift, Door, Push  (3/5 tasks)
  * CQL: Lift  (1/5 tasks)
  * SM: Lift, Door, Extraction, Push, Navigation  (5/5 tasks)
  * TM: Lift, Extraction, Push  (3/5 tasks)
  * OM: Lift, Door  (2/5 tasks)
  * CM: None  (0/5 tasks)
While I'd agree that SIG can improve sample efficiency in some tasks with some algorithms, the paper should discuss why it offers no improvement -- or worse performance -- in other task/algorithm combinations (e.g. CM + SIG and CQL + SIG).

* Figure 3 would be easier to read if curves for <algo> and <algo> + SIG had the same color but differen line styles (e.g. solid vs dashed). Also, figure labels are very tiny and difficult to read. Please use larger font sizes!


1. "This issue can be mitigated by improving the algorithm’s target updating or value estimation method
to encourage exploration, thereby improving sample efficiency." This statement is seemingly disconnected from the previous paragraph. Why should we immediately jump to target updates and value estimation to resolve issues with sample efficiency? What's the motivation?

2. "This module could verify the rationality of the imagined states" It's unclear what rationality means here.

[1] When to Trust Your Model: Model-Based Policy Optimization. https://arxiv.org/abs/1906.08253
[2] Deep Reinforcement Learning in a Handful of Trials using Probabilistic Dynamics Models. https://arxiv.org/abs/1805.12114

**Questions:**

See Weaknesses.

---

### Official Review · Reviewer_6ea1 · 2024-11-07

**Soundness:** 2
**Presentation:** 2
**Contribution:** 2
**Rating:** 3
**Confidence:** 4

**Summary:**

This paper proposes sample imagined generator (SIG), a method to synthesize imagined replay to increase the sample efficiency for the reinforcement learning algorithm. Experiments on 5 continuous control tasks proved SIG can improve the sample efficiency of 5 off-policy algorithms.

**Strengths:**

1. The writing and presentation is mostly clear.
2. The experiments study both off-policy RL settings and offline-to-online RL settings on 5 off-policy or offline RL algorithms.

**Weaknesses:**

1. Experiments need improvements

    1.1. Important baselines are missing. The author needs to compare SIG against SYNTHER [1], a method focuses on using imagined replay to improve RL's sample efficiency.


    1.2. The author needs to compare SIG against REDQ [2], and see if REDQ can be further improved by SIG


    1.3. Some claims are not rigorous or have factual errors. The baseline selection is very confusing. "we combine the state-of-the-art MCAC with SAC, TD3, GAE, OEFD, CQL, ... a total of 10 off-policy algorithms". First, "GQE enhances training stability based on SAC by incorporating a sophisticated reward estimation", GQE is more commonly called GAE instead, and it was published before SAC. It is not an off-policy algorithm.

    CQL is an *offline RL* algorithm. OEFD is based on DDPG (DDPG is an off-policy algorithm), but OEFD is more of an algorithm to leverage demonstrations (with state reset, hindsight, q filter, etc). Also combining them with MCAC will not make 10 off-policy algorithms. Furthermore, RLPD [3] is considered the current state-of-the-art approach rather than MCAC.


    1.4. The plots in Figure 3 are hard to read. For example, if you make SAC / SAC+SIG in the same color, but in solid / dash lines (same change for other algorithms), it may be better for the readers to tell (1) if SIG is improving (2) how different the performance across algorithms.

2. The selection about $H_{sig}$, $R_{sig}$, and $T_{sig}$ and other hyperparameters are all heuristic-based or a little bit arbitrary. More ablation studies are needed to test if they are sensitive to hyperparameter choices, e.g. $K$, $f_{less}$, $r_{max}$

[1] Lu et al., Synthetic experience replay

[2] Chen et al., Randomized Ensembled Double Q-Learning: Learning Fast Without a Model

[3] Ball et al., Efficient Online Reinforcement Learning with Offline Data

**Questions:**

1. Why there's a noise in the actual state transfer? Is $\Delta$ an exploration noise? Is the agent taking action $a_t + \Delta$?  If so, why the next state is computed based on $a_t$ but not the action the agent has taken?
2. Objective 13: what's $\mu_t$ and $\sigma_t$? how are they computed? where does $t$ come from? where is $i$?
3. Objective 14: what's $\mu(k)$ and $\sigma(k)$?
4. Pseudo code line 13: How are $\mu_t$ and $\sigma_t$ computed? By using all the $L_{ssg}$ so far, or in the inner loop?
5. The ablation is conducted only on the fixed switch. Does fixed length or sample ratio also hurt the performance a lot?

---

### Meta-Review · Area_Chair_J81R · 2024-12-16

**Metareview:**

This paper proposes an approach for generating synthetic data to train agents with reinforcement learning. The authors show improvements across continuous control tasks. The reviewers highlighted a series of weaknesses, in particular a lack of comparison vs. Synthetic Experience Replay and model based approaches, the natural baselines. Thus, the paper does not meet the bar for acceptance since there is no reasonable comparison. If indeed it does outperform these other approaches then it could be a strong submission to a future venue.

**Additional Comments On Reviewer Discussion:**

There was no discussion.

---

### Decision · Program_Chairs · 2025-01-22

Reject